# Preparation and Application of Efficient Biobased Carbon Adsorbents Prepared from Spruce Bark Residues for Efficient Removal of Reactive Dyes and Colors from Synthetic Effluents

**Glaydson Simões dos Reis** [1,*], **Sylvia H. Larsson** [1], **Mikael Thyrel** [1], **Tung Ngoc Pham** [2], **Eder Claudio Lima** [3], **Helinando Pequeno de Oliveira** [4] and **Guilherme L. Dotto** [5]

1 Biomass Technology Centre, Department of Forest Biomaterials and Technology, Swedish University of Agricultural Sciences, SE-901 83 Umeå, Sweden; sylvia.larsson@slu.se (S.H.L.); mikael.thyrel@slu.se (M.T.)
2 Faculty of Chemistry, The University of Danang-University of Science and Technology, 54 Nguyen Luong Bang, Lien Chieu, Da Nang 550000, Vietnam; pntung@dut.udn.vn
3 Institute of Chemistry, Federal University of Rio Grande do Sul (UFRGS), Av. Bento Gonçalves 9500, Porto Alegre 91501-970, RS, Brazil; profederlima@gmail.com
4 Institute of Materials Science, Federal University of Sao Francisco Valley, Juazeiro 48920-310, BA, Brazil; helinando.oliveira@univasf.edu.br
5 Chemical Engineering Department, Federal University of Santa Maria (UFSM), Santa Maria 97105-900, RS, Brazil; guilherme_dotto@yahoo.com.br
* Correspondence: glaydsonambiental@gmail.com or glaydson.simoes.dos.reis@slu.se

**Abstract:** Biobased carbon materials (BBC) obtained from Norway spruce (Picea abies Karst.) bark was produced by single-step chemical activation with $ZnCl_2$ or KOH, and pyrolysis at 800 °C for one hour. The chemical activation reagent had a significant impact on the properties of the BBCs. KOH-biobased carbon material (KOH-BBC) had a higher specific surface area ($S_{BET}$), equal to 1067 m² g⁻¹, larger pore volume (0.558 cm³ g⁻¹), more mesopores, and a more hydrophilic surface than $ZnCl_2$-BBC. However, the carbon yield for KOH-BBC was 63% lower than for $ZnCl_2$-BBC. Batch adsorption experiments were performed to evaluate the ability of the two BBCs to remove two dyes, reactive orange 16 (RO-16) and reactive blue 4 (RB-4), and treat synthetic effluents. The general order model was most suitable for modeling the adsorption kinetics of both dyes and BBCs. The equilibrium parameters at 22 °C were calculated using the Liu model. Upon adsorption of RO-16, $Q_{max}$ was 90.1 mg g⁻¹ for $ZnCl_2$-BBC and 354.8 mg g⁻¹ for KOH-BBC. With RB-4, Qmax was 332.9 mg g⁻¹ for $ZnCl_2$-BBC and 582.5 mg g⁻¹ for KOH-BBC. Based on characterization and experimental data, it was suggested that electrostatic interactions and hydrogen bonds between BBCs and RO-16 and RB-4 dyes played the most crucial role in the adsorption process. The biobased carbon materials showed high efficiency for removing RO-16 and RB-4, comparable to the best examples from the literature. Additionally, both the KOH- and $ZnCl_2$-BBC showed a high ability to purify two synthetic effluents, but the KOH-BBC was superior.

**Keywords:** biobased carbon materials; meso- and microporous carbons; dye adsorption; chemical adsorption; electrostatic interactions

## 1. Introduction

Biomass is a renewable and widespread resource that, if utilized sustainably, can help to reduce the emission of carbon dioxide that directly affects global warming [1]. Agricultural and forestry residues and by-products from biobased industries can be used as feedstock for energy production and material applications to replace fossil fuel sources [2,3].

Norway spruce (*Picea abies* (L.) Karst.) is one of the most common and economically valuable trees for the European forest industry as it is widely distributed from central to boreal and eastern Europe [4]. Together with pine and birch, spruce is the most common tree species in Sweden; these three combined comprise more than 90% of standing volume [4].

The Swedish annual forest harvest amounts to approximately 90 Mm$^3$ standing volume [5], and they are economically very important for sawmill and paper and pulp industries. However, in the production of sawn timber, pulp, and paper, only the stem wood is used—the remaining components can be considered industrial by-products. Around 10–15% of the feedstock volume delivered to the forest industries consists of bark currently mainly utilized as fuel and other low-value applications [5,6]. Consequently, research to employ bark as a precursor for value-added and eco-friendly material products is motivated.

Using biomass to produce biobased carbon materials (BBC) such as biochar (BC), activated carbon (AC), carbon composite materials (CCM) is an application with great potential. It reduces fossil carbon use and can provide new types of functionalities [6–14]. BBC is the oldest, most common, and efficient material for removing pollutants from aqueous media [7–13]. Besides its chemical stability, surface functionalities, high porosity, and specific surface area are essential characteristics for efficient application in the adsorption process [8–16]. However, high-purity activated carbons are expensive; therefore, the use of other biobased carbon materials can be explored as adsorbents for the removal of pollutants and micropollutants [8,10–16].

Adsorption is seen as one of the most suitable treatment methods for tackling pollutants from contaminated water and wastewaters due to its simple operating conditions, high efficiency, and low-cost employment. To design a very efficient adsorption process, the BBC must be prepared to achieve suitable properties.

The BBC properties are highly dependent on pyrolysis conditions and activation methods [14,17]. For instance, chemical activation can create carbon materials with ultra-high BET surface area ($S_{BET}$) and porosities because of extensive micro and mesoporosity development. Each pore structure has a specific role in the adsorption process, e.g., the micropore structure contributes significantly to the $S_{BET}$ values and the adsorption of small-sized contaminants (e.g., metallic species and small organic molecules) [13,17]. Mesopores are essential as vectors to the surface areas within the carbon material particle, and their respective quantities are primarily dependent on the pyrolysis conditions and activation method. The mesopore structure is vital for larger-molecule adsorption, which is the case for dyes and colored effluents.

It is estimated that over 10,000 different dyes and pigments are used in the food, leather, cosmetics, and textile industries, e.g., only the textile industry consumes up to 200,000 tons of dyes yearly, thereby generating large amounts of colored effluents [18]. These colored effluents are, if not adequately treated, discharged into the environment, where they are potentially harmful to the aquatic systems and ecosystem integrity. Besides, many dyes are reported as mutagenic and carcinogenic [19]. Therefore, these effluents must be treated before their discharge into the environment, and the adsorption process using biomass-activated carbon is one the most suitable treatment process [12–16].

The purpose of this study was to investigate the potential of spruce bark residues as a precursor to producing efficient carbon-based materials by pyrolysis, using KOH (KOH-BBC) and ZnCl$_2$ (ZnCl$_2$-BBC) as chemical activators. The effect of the chemical reagents on the BBC characteristics such as morphology, specific surface area and porosity, surface chemistry, BBC composition, hydrophobicity index, carbon yield, and adsorption of two dyes and different synthetic effluents were evaluated.

## 2. Materials and Methods

### 2.1. Preparation of BBCs

Norway spruce (*Picea abies* Karst.) bark was delivered from a pulp and paper mill in northeast Sweden and prepared at the Biomass Technology Centre (BTC), Swedish University of Agricultural Sciences, Umeå, Sweden. The wet bark was dried in a custom-made plane drier at 40 °C, shredded with a screen size of 15 mm (Lindner Micromat 2000, Lindner-Recyclingtech Gmbh), hammer-milled with a screen size of 4 mm (Bühler DFZK 1, Bühlergroup), representatively sampled according to ISO 18135:2017, and

cutting-milled with a screen size of 200 μm using a Fritsch Pulverisette 14 mill (FRITSCH GmbH, Germany).

The pyrolysis was done in a single pyrolysis-step preparation according to a previously reported procedure [20–24]. First, 15.0 g of the spruce bark was mixed in a weight ratio of 1:1 with each chemical activation agent (KOH and $ZnCl_2$). During the mixing, about 40.0 mL of water was added to form homogeneous pastes. These two pastes were dried in an oven at 105 °C for 24 h. The dried pastes were placed in a metallic crucible and treated thermally in a conventional high-temperature oven under a nitrogen flow of 600 mL min$^{-1}$. They were heated from 20 to 800 °C at a rate of 10 °C min$^{-1}$ and held at 800 °C for 60 min. The oven was turned off to cool down the pyrolyzed samples while the nitrogen flow was kept, and when the temperature dropped to 200 °C, the nitrogen flow was shut off. The pyrolyzed materials were milled with a screen size of 200 μm and completely leached out by conventional leaching using 0.1 M HCl, under reflux, for 2 h for KOH-BBC and using 1.0 M HCl, under reflux, for 2 h for $ZnCl_2$-BBC [13,15–17]. The BBC preparation procedure is summarized in Figure 1.

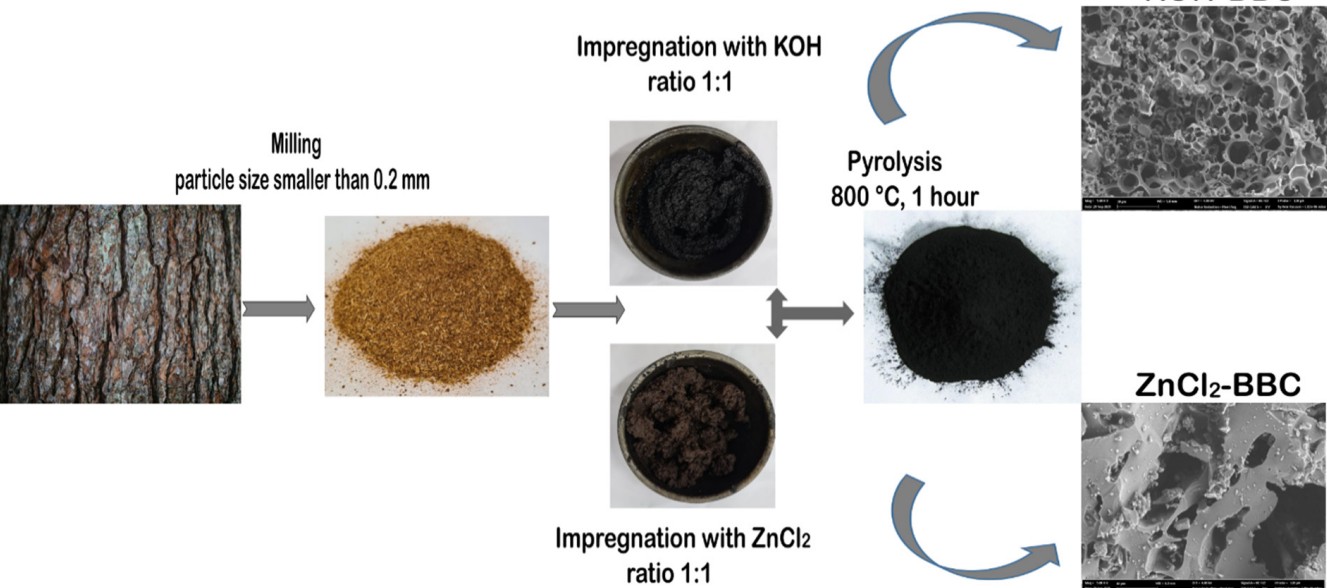

**Figure 1.** The BBC preparation procedure.

*2.2. BBC Characterization*

2.2.1. Textural Properties

The adsorbent's textural properties, especially for biobased carbon materials, are crucial for evaluating their applications and potential application efficiencies. Common BBCs are almost always heterogeneous, having an unknown range of pore sizes and a range of pore shapes, blocked and network pores [12,14,15].

The samples' surface morphology was observed with scanning electron microscopy (SEM) (55-VP, Supra, Zeiss), using an acceleration voltage of 20 kV and magnification ranging from 100 to 20,000.

$N_2$ adsorption/desorption isotherm analysis (Tristar 3000 apparatus, Micrometrics Instrument Corp., Norcross, GA, USA) was performed to quantify the porosity (by DFT method) and surface area (BET method). Before the analysis, the sample was degassed at 180 °C for 3 h in an $N_2$ atmosphere. The specific surface area was calculated in the relative pressure interval of 0.05–0.3 using the Brunauer–Emmett–Teller (BET) method [16,22]. Mesopore size and distribution were calculated by the Barrett–Joyner–Halenda (BJH) method from desorption curves while the micropore area values were calculated by the

t-plot method [16,20,22]. The percentage of the mesopore and micropore areas were calculated based on the $S_{BET}$ values [22].

2.2.2. Elemental Analysis, Yield (%), Raman Spectroscopy, and Zeta Potential

The elemental analysis was carried out to evaluate the volatiles and fixed carbon contents and quantify the elemental composition of the BBCs, respectively. The analysis was made using a CHN Perkin Elmer M CHNS/O Analyzer, model 2400.

The yield (%) was calculated from the dry matter quota of the biomass precursor after and before activation.

Raman spectroscopy is widely applied to characterize BBCs. It is applied to obtain structural information on the bulk of carbon materials. Raman spectra were recorded on a Renishaw inVia Raman spectrometer (Renishaw, Kingswood, UK) at 633 nm HeNe laser.

Zeta-potential was performed to obtain the charge (whether positive or negative) of the BBCs. It was determined at pH 7 using a potential analyzer (Zetasizer Nano ZS90, Malvern Panalytical, Malvern, UK).

2.2.3. Water Vapor Sorption and Hydrophobicity/Hydrophilicity

The BBCs' water vapor sorption isotherms and the hydrophobicity/hydrophilicity index (HI) are used to determine the properties of the BBC/water interface and the water molecules' ability to attach to the BBC surface, which both may influence dye adsorption.

The BBCs' $H_2O$ vapor adsorption isotherms were determined by dynamic vapor sorption (DVS Advantage, Surface Measurement Systems) at 25 °C, where RH was varied from 0 to 95% and back in 5% steps. The hydrophobicity/hydrophilicity index (HI) was performed according to a method previously reported in the literature [23]: 0.3 g of each BBC was placed into 5 mL beakers and inserted into plugged 1.5 L E-flasks with saturated atmosphere solvent vapor (water or n-heptane) using 80 mL of each solvent. The beakers were placed in the center of the E-flasks to avoid contact with the flask walls. After 24 h, the beakers were removed and weighed. The weight gained was used to calculate the maximum vapor adsorption.

*2.3. Dye Adsorption Analysis*

2.3.1. Batch Adsorption Studies

Aliquots of 20.00 mL of 30.00–1000.0 mg $L^{-1}$ of RB-4 and RO-16 were added to 50.0 mL Falcon flat tubes containing 30 mg (dosage of 1.5 g $L^{-1}$) of each BBC [20,24,25]. The Falcon tubes containing RB-4 or RO-16 and BBCs were agitated in a shaker model TE-240 between 0.1–12 h to obtain the kinetics data. Afterward, to separate the dyes and BBCs, the flasks were centrifuged. After adsorption, the residual solutions of RB-4 and RO-16 were quantified using a UV-Visible spectrophotometer (Shimadzu 1800) at a maximum wavelength of 595 and 494 nm, respectively.

The amount of RO-16 adsorbed by the BBCs and the percentage of removal were calculated using Equations (1) and (2), respectively [20,24,25]:

$$q = \frac{\left(C_0 - C_f\right)}{m} \cdot V \tag{1}$$

$$\% \; Removal = 100 \cdot \frac{\left(C_0 - C_f\right)}{C_0} \tag{2}$$

where $q$ is the amount of selected dye uptake by the BBCs (mg $g^{-1}$); $C_0$ is the initial dye concentration in contact with BBCs (mg $L^{-1}$), $C_f$ is the final concentration (mg $L^{-1}$) after adsorption, $V$ is the volume of dye solutions (L) in contact with the BBCs, and $m$ is the BBC mass (g).

### 2.3.2. Adsorption Kinetics and Equilibrium Analysis

Adsorption kinetics provides information on the adsorption rate, the adsorbent's performance, and the mass transfer mechanisms [20,23–25]. Knowing the adsorption kinetics is crucial for designing efficient adsorption systems.

The RO-16 adsorption kinetics of the KOH-BBC and ZnCl$_2$-BBC samples were evaluated at two initial concentrations: 500 and 700 mg L$^{-1}$. The suitability of different models for predicting the adsorption kinetics was assessed by analyzing R$^2$$_{adj}$ and SD values. Pseudo-first-order (PFO) model, pseudo-second-order (PSO) model, and general order models were used to evaluate the kinetic adsorption process [23–25].

The mathematical representations of pseudo-first-order, pseudo-second-order, and general order are shown in Equations (3)–(5), respectively.

$$q_t = q_e \cdot [1 - \exp(-k_1 \cdot t)] \tag{3}$$

$$q_t = \frac{k_2 \cdot q_e^2 \cdot t}{1 + q_e \cdot k_2 \cdot t} \tag{4}$$

$$q_t = q_e - \frac{q_e}{\left[ k_N \cdot (q_e)^{n-1} \cdot t \cdot (n-1) + 1 \right]^{1/(n-1)}} \tag{5}$$

Equilibrium isotherms are used to determine the adsorption affinity and dye removal mechanisms of the adsorption systems [8,23–25]. Each adsorption system (individual adsorbent material and adsorbate) has a unique isotherm, and the quantity of adsorbed adsorbate on an adsorbent depends on both the BBC's and the solution's properties. Therefore, equilibrium studies are mandatory to evaluate and establish adsorbent efficiency.

The equilibrium process was analyzed by Langmuir, Freundlich, and Liu's models. The fit quality was assessed through statistical indicators such as $R^2$, $R^2_{adj}$, and $SD$. See further details about these indicators in references [13,14,20,23].

Langmuir, Freundlich, and Liu's models are shown in Equations (6)–(8), respectively.

$$q_e = \frac{Q_{max} . K_L . C_e}{1 + K_L . C_e} \tag{6}$$

$$q_e = K_F . C_e^{1/nF} \tag{7}$$

$$q_e = \frac{Q_{max} . (K_g . C_e)^{nL}}{1 + (K_g . C_e)^{nL}} \tag{8}$$

Detailed information about all these equations can be found in the literature [10,23,24].

### 2.3.3. Preparation of the Dyeing Synthetic Effluents

De-ionized water was used for the preparation of all solutions used in the dye adsorption experiments. RB-4 ($C_{23}H_{14}N_6Cl_2O_8S_2$) and RO-16 ($C_{20}H_{17}N_3O_{10}S_3Na_2$) were obtained from Sigma Aldrich, Sweden. The stock solution was prepared by dissolving the dye in distilled water to 2.00 g L$^{-1}$. Working solutions were obtained by diluting the dye stock solution to the required concentrations without adjusting the pH.

Synthetic effluents with different compositions (see Table 1) were prepared to test the BBCs' applicability for treating real effluents.

### 2.3.4. Analytical Control of the Measurements and Statistical Evaluation of Nonlinear Models

The adsorption equations were fitted using the nonlinear approach obtained by the Simplex method and the successive interactions of the Levenberg–Marquardt algorithm [10,15–17,21]. This fitting was acquired by the nonlinear fitting facilities of the Microcal Origin 2020 software, and they were used to fit the kinetic and equilibrium data. The determination coefficient ($R^2$), adjusted determination coefficient ($R^2_{adj}$), and the

standard deviation of the residues (SD) were employed to analyze the suitability of the models [10,15–17,21–25].

**Table 1.** Effluent compositions and concentrations.

| Compounds | Concentration (mg L$^{-1}$) | | $\lambda_{max}$ (nm) |
|---|---|---|---|
| Effluent | A | B | |
| RO-16 | 50 | 50 | 494 |
| RB- 4 | 50 | 50 | 595 |
| Methylene Blue | 50 | 50 | 668 |
| Bismarck Brown | 50 | - | 468 |
| Crystal Violet | 50 | - | 590 |
| Methyl Red | - | 50 | 507 |
| Methyl Orange | - | 50 | 522 |
| Phenol Red | - | 50 | 550 |
| Sodium Dodecyl | 25 | 25 | - |
| Sodium sulfate | 25 | 25 | - |
| Ammonium chloride | 20 | 25 | - |
| Sodium acetate | 20 | 25 | - |
| pH | 5.1 | 4.9 | - |

Residual standard deviation measures the difference between the theoretical and experimental amounts of dyes removed from solutions. The $R^2$, $R^2_{adj}$, and $SD$ are given in Equations (9)–(11), respectively [21–25].

$$R^2 = \left( \frac{\sum_i^n \left( q_{i,exp} - \bar{q}_{i,exp} \right)^2 - \sum_i^n \left( q_{i,\,exp} - q_{i,\,model} \right)^2}{\sum_i^n \left( q_{i,exp} - \bar{q}_{i,exp} \right)^2} \right) \tag{9}$$

$$R^2_{adj} = 1 - \left( 1 - R^2 \right) \cdot \left( \frac{n-1}{n-p-1} \right) \tag{10}$$

$$SD = \sqrt{ \left( \frac{1}{n-p} \right) \cdot \sum_i^n \left( q_{i,\,exp} - q_{i,\,model} \right)^2 } \tag{11}$$

where $q_{i,model}$ represents the individual theoretical $q$ value predicted by the model. $q_{i,exp}$ represents each experimental $q$ value. $\bar{q}_{exp}$ is the average of the experimental $q$ values. $n$ and $p$ represent the number of experiments and model parameters, respectively.

## 3. Results and Discussion

### 3.1. BBC Characteristics

3.1.1. Textural Properties and Porosity

The SEM images show remarkable differences between both microstructures (see Figure 2). The ZnCl$_2$-BBC has a dense structure, with more elongated cavities and holes of different sizes and shapes (Figure 2A,C) that should have been formed during the leaching step with 6.0 mol L$^{-1}$ HCl. Additionally, it is observed that ZnCl$_2$-BBC presents a rough surface.

KOH-BBC (Figure 2B,D) presents ordered macropore structure and holes with lower diameter in its surface, which should be attributed to the lower concentration of HCl (1.0 mol L$^{-1}$) used in the leaching step. Both BBC presents irregular particle size and rough surface.

The ZnCl$_2$-BBC was prepared by catalyzed dehydration and elimination of carbonyl and hydroxyl groups during the heat treatment [26,27], and that the ZnCl$_2$ (due to its low melting point at 290 °C and the boiling point at 732 °C) is fused into the biomass matrix, thereby creating a denser structure and a microporous network [27,28]. On the other hand, KOH activation provokes the breakage of C–O–C and C–C bonds, creating

pores and well-developed porosity [26]. Additonally, an uneven distribution of KOH in the bark matrix can promote hyperactivation during the pyrolysis process, resulting in pore wall demolition and widening of the micropores into mesopores [26]. These structural transformations are beneficial for the BBC's physical adsorption of RB-4 and RO-16 and effluents treatment.

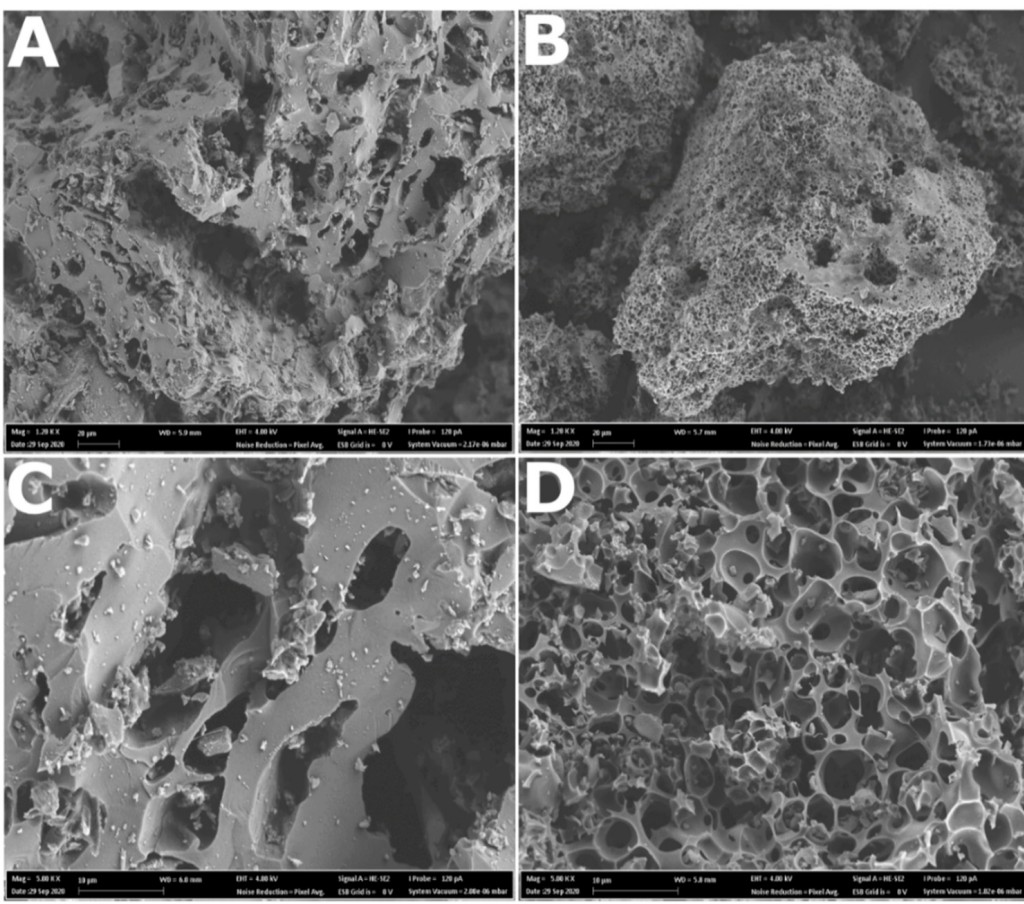

**Figure 2.** SEM images of BBCs: (**A**) $ZnCl_2$-BBC at 1.2 K of magnification, (**B**) KOH-BBC at 1.2 K of magnification, (**C**) $ZnCl_2$-BBC at 5 K of magnification, (**D**) KOH-BBC at 5 K of magnification.

The $N_2$ isotherms for $ZnCl_2$-BBC and KOH-BBC (Figure 3) can be ascribed to a type I isotherm. A type I isotherm (also mentioned as Langmuir isotherm) is typical for microporous materials (with pore diameter <2 nm) [16]. Higher amounts of $N_2$ are adsorbed at low relative pressures for microporous materials, and when it is close to 1, the curve may reach a limiting value or rise if larger pores are present [16].

Although both BBCs exhibited a Type I isotherm, the adsorbed $N_2$ volumes differed significantly (Figure 3). The KOH-treated BBCs had an almost 30% higher $S_{BET}$ (1067 m$^2$ g$^{-1}$) than the $ZnCl_2$-BBC (754 m$^2$ g$^{-1}$) (Table 2). The external surface area and the micropore and mesopore volumes agree with these results. Hence, it can be concluded that KOH-activation produced a BBC with better textural properties and better adsorption performance than $ZnCl_2$-activation. The percentage of mesopores in KOH-BBC (49.29%) is higher when compared with the $ZnCl_2$-BBC sample (43.51%), while the share of micropores is higher in $ZnCl_2$-BBC (56.49%) when compared to KOH-BBC (50.71%) (See Table 2). However, both micro and mesoporous materials are highly efficient to adsorb organic molecules with small sizes and, therefore, suitable for adsorption of RB-4 (size of 1.59 nm) and RO-16 (size of 1.68 nm).

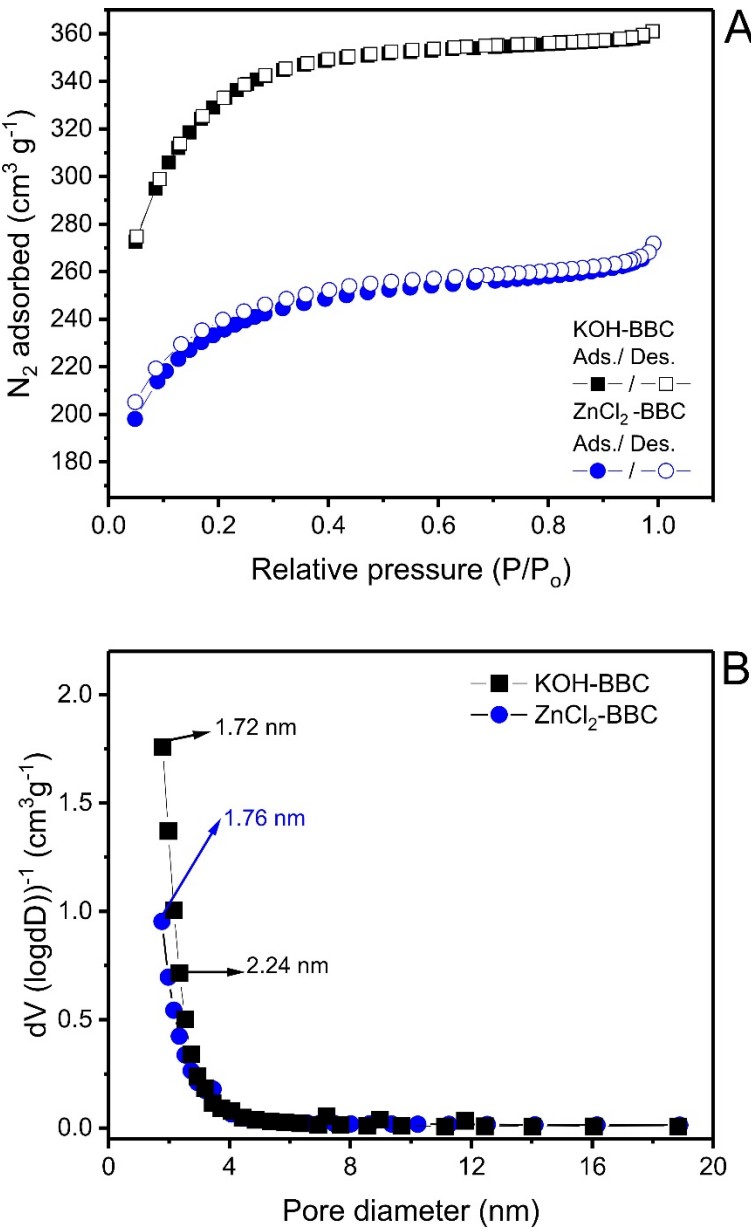

**Figure 3.** N$_2$ isotherm curves for ZnCl$_2$-BBC and KOH-BBC (**A**) and pore distribution curves (**B**).

**Table 2.** Textural properties of the activated carbons.

| Samples | ZnCl$_2$-BBC | KOH-BBC |
|---|---|---|
| Parameters | | |
| S$_{BET}$ (m$^2$ g$^{-1}$) | 754 | 1067 |
| External surface area (m$^2$ g$^{-1}$) | 328 | 526 |
| % of mesopore area (%) | 43.51 | 49.29 |
| t-plot Micropore area (m$^2$ g$^{-1}$) | 425.8 | 541.2 |
| % of micropore area (%) | 56.49 | 50.71 |
| Total pore volume (cm$^3$ g$^{-1}$) | 0.4205 | 0.5585 |
| t-plot micropore volume (cm$^3$ g$^{-1}$) | 0.2172 | 0.2776 |
| % of micropore volume (%) | 51.65 | 49.70 |
| Volume of mesopores (cm$^3$ g$^{-1}$) | 0.2033 | 0.2809 |
| Average pore size (nm) | 2.231 | 2.093 |

The pore size distributions derived from the BJH plots of both BBC samples are displayed in Figure 3B. The chemical activation seemed to affect the pore structure of the BBC samples. BBC-KOH showed a much higher distribution of the pores in the range of large micropores or small mesopores, 1.72–2.24 nm (see the line with squares). According to the BJH plots, both samples possess large quantities of micropores and homogeneous and small mesopores. The creation of large micropores and small mesopores enhanced the BBC-KOH sample, which is in good agreement with the porosity data.

Literature data reveals significant variance in $S_{BET}$ values depending on the type of biomass and preparation conditions (Table 3). For instance, Sipola et al. [8] prepared activated carbon from scots pine (*Pinus sylvestrus*) and spruce (*Picea* spp.) barks for wastewater purification and found specific surface areas ranging from 200 to 600 $m^2 g^{-1}$. In another work [9], the spruce bark porous materials were produced and employed in methylene blue dye adsorption. The materials presented $S_{BET}$ ranging from 351 to 1275 $m^2 g^{-1}$ and were successfully employed in the dye removal from aqueous solutions. In addition, a specifically high $S_{BET}$ (2330 $m^2 g^{-1}$) was achieved with rice plant residue as a biomass precursor. However, in that case, a highly complex preparation procedure was required: First, pre-carbonization at 500 °C for one hour followed by NaOH washing; Secondly, BBC was mixed with KOH at a ratio of 1:4 (biomass: KOH) and pyrolyzed at 800 °C for 30 min and then, followed by HCl washing to remove the inorganic compounds. Consequently, due to the cumbersome procedure, the high $S_{BET}$ comes with a high cost. The $S_{BET}$ values of the $ZnCl_2^-$ and KOH-activated Norway spruce bark BBCs are comparable with BBCs from several other biomass precursors, but in this case, the manufacturing method is simple, and the feedstock material highly available and cheap.

**Table 3.** Comparison of KOH-BBC and ZnCl2-BBC preparation methods and $S_{BET}$ for a variety of biomass precursors.

| Adsorbent | Activation Reagent | Preparation Conditions | $S_{BET}$ ($m^2 g^{-1}$) | Ref. |
|---|---|---|---|---|
| Scots pine bark | Steam + $N_2$ | Firstly, the biomass was carbonized using slow pyrolysis at 475 °C for 3 h. Afterward, heated at 800 °C for 3.5 h under steam activation [steam (30 and 40%) + $N_2$ (66 and 300 L/h)]. | 539–603 | [8] |
| Norway spruce bark | Steam + $N_2$ | Firstly, the biomass was carbonized using slow pyrolysis at 475 °C for 3 h. Afterward, heated at 800 °C for 3.5 h under steam activation [steam (30 and 40%) + $N_2$ (66 and 300 L/h)]. | 187– 369 | [8] |
| Norway spruce bark | Steam + $N_2$ | The biomass was heated at 600 °C for 2 h under steam activation (steam + $N_2$). | 351 | [9] |
| Norway spruce bark | $ZnCl_2$ | A mixture of $ZnCl_2$ and biomass powder at ratio 2.0:1.0 ($ZnCl_2$:biomass) and pyrolyzed at 600 °C for 2 h. Afterward, it was washed with HCl to remove the inorganic compounds. | 1495 | [9] |
| Tea leave residue | KOH | A mixture of KOH and tea powder (2:1) and pyrolyzed at 900 °C for 60 min. Afterward, it was washed with HCl to remove the potassium compounds and further pyrolyzed at 1200 °C. | 912 | [28] |
| Palm shell | KOH+ $ZnCl_2$ | Pre-carbonization of biomass at 400 °C for 2 h. Afterward, a mixture of biomass and both KOH (75%) and $ZnCl_2$ (25%) at the final ratio of biomass: chemical activator 1:4. The mixture was then pyrolyzed at 850 °C for 1 h and washed with HCl. | 1295 | [29] |
| Garlic peel | KOH | First, it was hydro-carbonized and then chemically activated by KOH (ratio 2:1, KOH: biomass) and pyrolyzed at 600 °C at 4 °C/min under $N_2$ flow for 2 h. | 947 | [30] |

**Table 3.** *Cont.*

| Adsorbent | Activation Reagent | Preparation Conditions | $S_{BET}$ $(m^2\,g^{-1})$ | Ref. |
|---|---|---|---|---|
| Rice plants | KOH | The biomass was Pre-carbonized at 500 °C for 1 h, followed by NaOH washing. Afterward, the pyrolyzed BBC was mixed with KOH at ratio 1:4. The mixture was then pyrolyzed at 800 °C for 30 min and then washed with HCl. | 2330 | [31] |
| Brazil nutshells | $ZnCl_2$ | A mixture of $ZnCl_2$ and biomass powder at ratio 1.5:1.0 ($ZnCl_2$:biomass) and pyrolyzed at 600 °C for 30 min. Afterward, it was washed with 6.0 M HCl to remove the inorganic compounds. | 1457 | [32] |
| Sewage sludge | $ZnCl_2$ | A mixture of $ZnCl_2$ and biomass powder at ratio 0.5:1.0 ($ZnCl_2$:biomass) and pyrolyzed at 500 °C for 15 min. Afterward, it was washed with HCl to remove the inorganic compounds. | 679 | [33] |
| Coconut shell | $ZnCl_2$ | Blending coconut shell powder and $ZnCl_2$ at ratio 1:3 in 50 mL of 3 M $FeCl_3$ solution. Afterward, heated at 900 °C for 1 h under an inert atmosphere. Afterward, it was washed with HCl to remove the inorganic compounds. | 1874 | [34] |
| Norway spruce bark | $ZnCl_2$ | $ZnCl_2$ and biomass powder mixture at ratio 1.0:1.0 ($ZnCl_2$:biomass) and pyrolyzed at 800 °C for 60 min. Afterward, it was washed with 6.0 M HCl to remove the inorganic compounds. | 754 | This work |
| Norway spruce bark | KOH | A mixture of KOH and biomass powder at ratio 1.0:1.0 (KOH: biomass) and pyrolyzed at 800 °C for 60 min. Afterward, it was washed with 1.0 M HCl to remove the inorganic compounds. | 1067 | This work |

3.1.2. Elemental Analysis, Carbon Yield, Raman Spectroscopy, Zeta-Potential, and FTIR

The carbon content of the spruce bark $ZnCl_2$- and KOH-activated BBCs was 94.8% and 91.6%, respectively (see Table 4). These values are very high compared to literature; Correa et al. [35] produced several BBCs from different biomasses, and the carbon content varied from 76.9 to 87.8%, while Duan et al. [36] obtained 82.66% of carbon content in BBC made from coconut shells. High carbon content can reflect good adsorption efficiency because hydrophobic interactions of the aromatics of BBC can interact with organic molecules. In addition, high carbon content means less ash content, and ashes in the BBC reduce $S_{BET}$ and functional groups, which hinder the adsorption process. Concerning the oxygen content, KOH-BBC presented higher content when compared to $ZnCl_2$-BBC; this can positively influence the carbons' hydrophilicity index and the water/dye adsorption behavior [35].

$$HI = \frac{\frac{amount\ of\ water\ vapor\ (mg)}{mass\ of\ BBC\ (g)}}{\frac{amount\ of\ h-heptane\ vapor\ (mg)}{mass}of\ BBC\ (g)} \tag{12}$$

The *BBC* yield from pyrolysis and KOH activation was approximately one-third of the $ZnCl_2$ treatment (Table 4). This result indicates a strong reaction between bark and KOH during the pyrolysis process. Via breakage of C–O–C and C–C bonds, KOH can play a catalytic role in the material's volatilization, leading to a low carbon yield [37]. Impregnation with $ZnCl_2$ results in degradation of the cellulosic material that, combined with the dehydration during carbonization, leads to charring and aromatization of the carbon skeleton. These pyrolytic conditions inhibit the formation of tar and reduce mass loss [38]. As a result, BBC production by $ZnCl_2$ activation generally provides higher yields than when using other chemical reagents [38].

**Table 4.** Properties and elementary analysis of activated carbons.

| Samples | ZnCl$_2$-BBC | KOH-BBC |
|---|---|---|
| Parameters | | |
| HI (H$_2$O/n-heptane) | 1.19 | 1.29 |
| Zeta potential (mV) | −19.4 | −20.5 |
| pH | 5.1 | 6.0 |
| Carbon content (%) | 94.8 | 91.6 |
| Nitrogen content (%) | 0.51 | 0.29 |
| Hydrogen (%) | 1.2 | 1.6 |
| Oxygen (%) | 2.5 | 5.3 |
| Ash (%) | 0.99 | 1.21 |
| BBC yield (%) | 38.1 | 14.2 |

The Raman spectra of the ZnCl$_2$- and KOH-BBCs are shown in Figure 4. The D and G bands indicate the degree of defective structure and the activated carbons' graphitization, respectively [28,30]. These bands' position, area, and intensity can also show differences in the structural characteristics [34,39]. Both samples' D- and G-bands are located at around 1340 and 1593 cm$^{-1}$, corresponding to the defect/disorder-induced structures in the BBCs' graphite layers and the vibration of sp2-bonded carbon atoms in a two-dimensional hexagonal lattice, respectively [28,34]. The relative strength intensity (I$_D$/I$_G$) represents the degree of defect in the BBCs—higher values indicate more defects [30]. The obtained I$_D$/I$_G$ values were 1.1 and 0.99 for ZnCl$_2$-BBC and KOH-BBC, respectively, i.e., the graphitization level in the KOH-BBC was slightly higher than in the ZnCl$_2$-BBC [28,30].

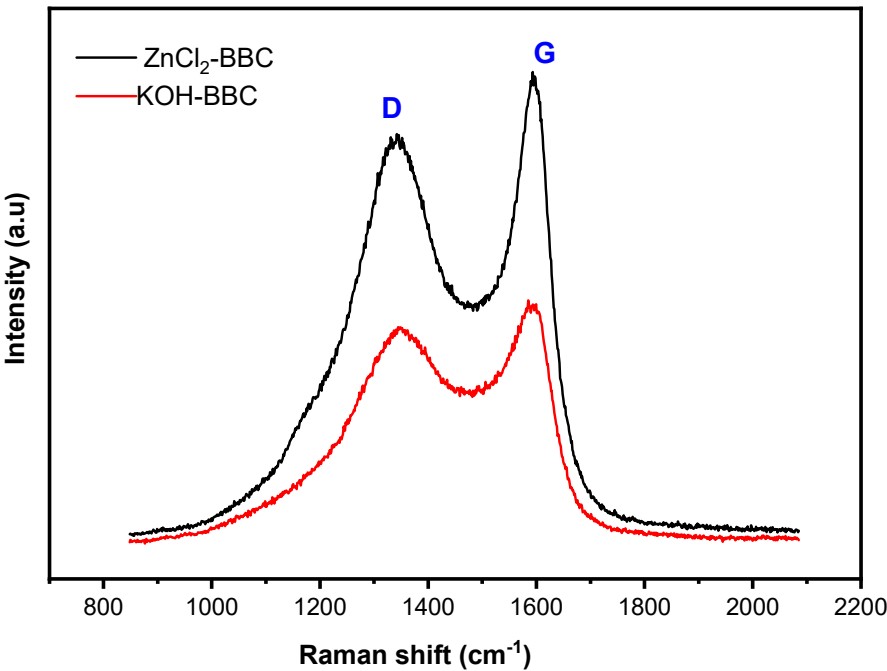

**Figure 4.** Raman spectra of BBC samples.

The Zeta-potential of both BBCs were negative, with a slightly higher value for the KOH-BBC (see Table 4). The negative charging comes from COO–, –COH–, and –OH– functionalities that can positively affect the adsorption process [40].

FTIR was employed to identify the presence of the functional groups on BBCs samples. The FTIR spectra of the BBC samples are presented in Figure 5. It is possible to identify that the different chemical treatments affected the chemical functionalities on the BBCs. In KOH-BBC, the presence of peaks in between 4000–3600 cm$^{-1}$ represents the O–H stretching vibration in carboxyl and phenol groups [10–12,15]. The sample treated with

KOH also exhibited a sharper and broader transmittance band at 3410–3535 cm$^{-1}$ when compared with the ZnCl$_2$-treated sample, which is assigned to the O–H stretching mode of hydroxyl groups and adsorbed water [11,12,15]. The peaks at 2948 cm$^{-1}$ (asymmetric) and 2875 cm$^{-1}$ (symmetric) are related to the CH– stretching and appeared only in the sample treated with ZnCl$_2$. A new peak at 2373 cm$^{-1}$ is observed only in KOH-BBC, which is assigned to hydrogen-bonded OH. The peaks at around 1542–1574 cm$^{-1}$ are assigned to the asymmetric stretching of O=C of carboxylates. The band at 1138–1160 cm$^{-1}$ are related to CO– of alcohols, and at around 963–1009 cm$^{-1}$ to the OCC—a stretch of an ester is identified [10–12,15]. These functional groups on BBCs surfaces are often related to a good adsorption efficiency process [10–12,15].

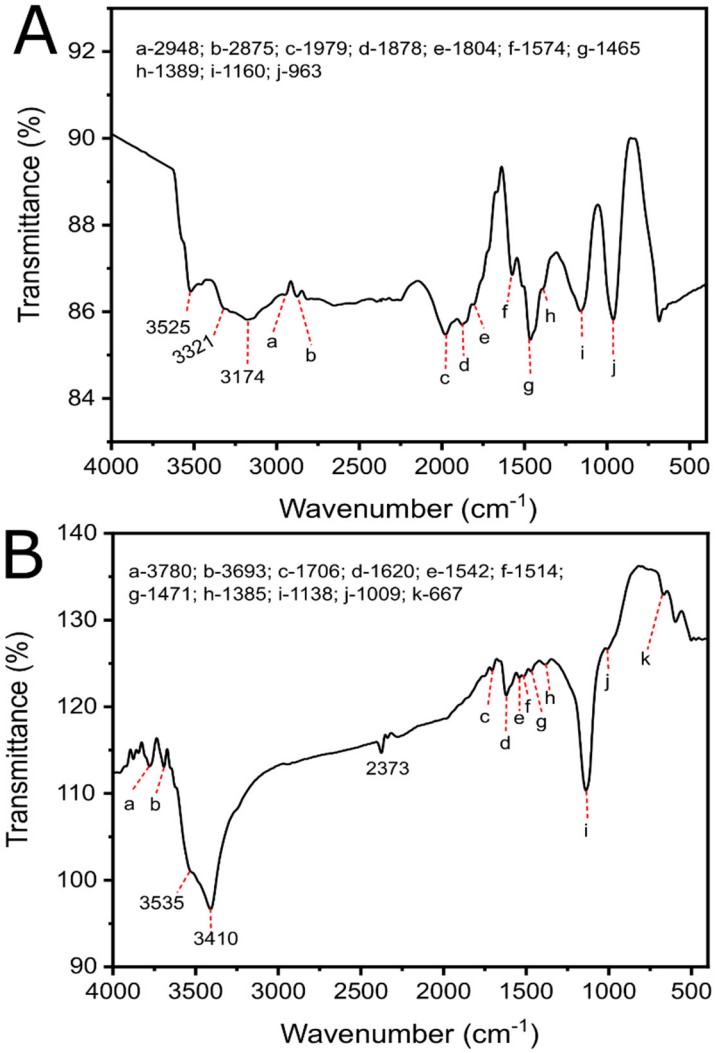

**Figure 5.** FTIR spectra of BBC samples (**A**) ZnCl$_2$-BBC and (**B**) KOH-BBC.

### 3.1.3. Water Vapor Adsorption Isotherms, Hydrophilicity Index (HI)

Water vapor adsorption isotherms for both BBCs are shown in Figure 6. According to the IUPAC classification [41], both isotherms are very close to type V, characterized by low levels of water uptake at low relative pressures and the presence of a hysteresis loop over the majority of the pressure range. Adsorption of water vapor was higher for KOH-BBC than for ZnCl$_2$-BBC (see Table 4), indicating a more hydrophilic surface for KOH-BBC than ZnCl$_2$-BBC.

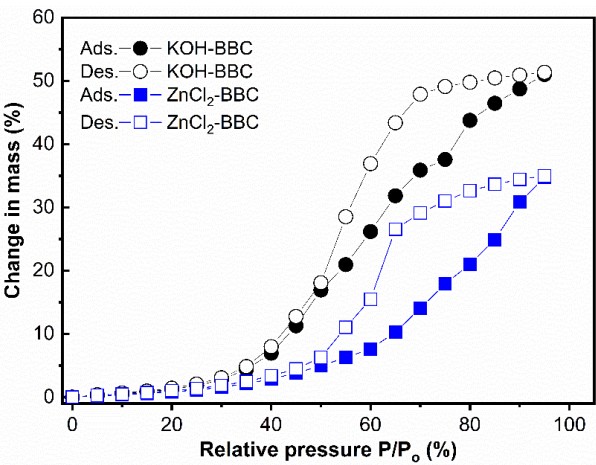

**Figure 6.** Water sorption isotherms for KOH-BBC and ZnCl$_2$-BBC samples at 25 °C.

The N$_2$ and H$_2$O isotherms differ both in type and shape. Although there is a nonexisting correlation between these two techniques, it is worth pointing out that N$_2$ adsorption generated type I isotherms, while H$_2$O adsorption yields isotherms of type V. Different on isotherm curves may be because the process is complex and does not depend only on the porosity. The adsorption of water vapor on biomass materials is known to be dependent on surface chemistry. BBC materials have plenty of surface functional groups, which initiate predominant water adsorption through the hydrogen bonding between a water molecule and surface functional groups.

### 3.2. Dye Adsorption Analysis
Adsorption Kinetics

The kinetic curves and their parameters are shown in Figure 7 and Table 5, respectively.

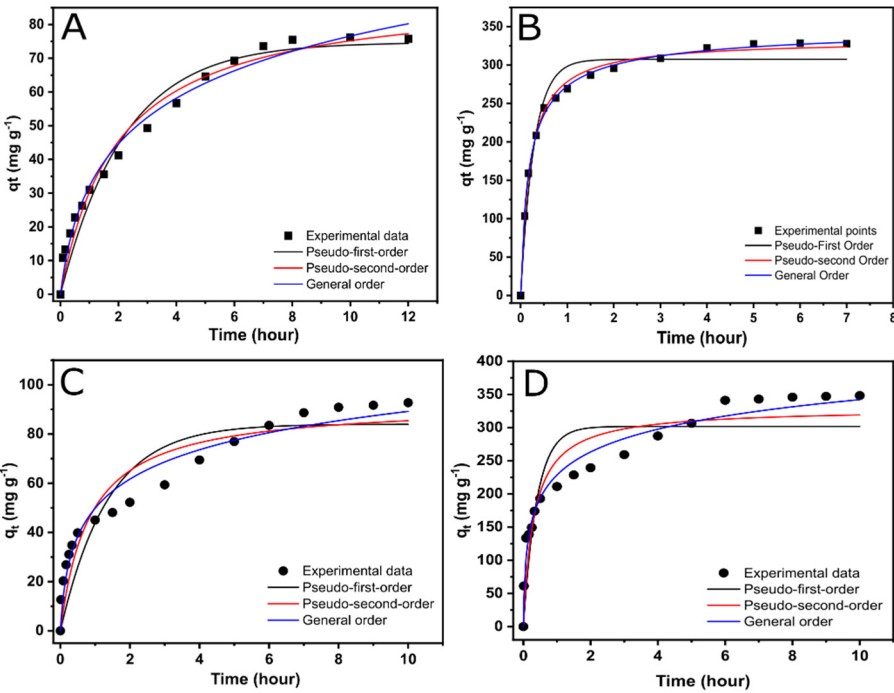

**Figure 7.** Kinetics of adsorption curves for uptake of RO-16 onto ZnCl$_2$-BBC (**A**) and uptake of RO-16 onto KOH-BBC (**B**), uptake of RB-4 onto ZnCl$_2$-BBC (**C**), uptake of RB-4 onto KOH-BBC (**D**). Initial pH of 5.5 and 4.0 for RO-16 and RB-4, respectively, the adsorbent dosage of 1.5 g L$^{-1}$. The temperature was 22 °C.

**Table 5.** Kinetic parameters of RO-16 and RB-4 adsorption onto the BBC samples.

| Model | RO-16 Initial Concentration (1000 mg L$^{-1}$) | | RB-4 Initial Concentration (1000 mg L$^{-1}$) | |
|---|---|---|---|---|
| | ZnCl$_2$-BBC | KOH-BBC | ZnCl$_2$-BBC | KOH-BBC |
| Pseudo-first order | | | | |
| $q_1$ (mg g$^{-1}$) | 74.71 | 307.5 | 84.01 | 301.7 |
| $k_1$ (min$^{-1}$) | 0.4529 | 3.502 | 0.7381 | 2.489 |
| $R^2$ | 0.9639 | 0.9606 | 0.8502 | 0.8182 |
| $R^2_{adj}$ | 0.9614 | 0.9573 | 0.8408 | 0.8068 |
| $SD$ (mg g$^{-1}$) | 5.107 | 20.22 | 11.79 | 46.03 |
| Pseudo-second order | | | | |
| $q_2$ (mg g$^{-1}$) | 89.96 | 332.7 | 92.62 | 329.1 |
| $k_2$ (g mg$^{-1}$ min$^{-1}$) | 0.00566 | 0.01539 | 0.01277 | 0.009700 |
| $R^2$ | 0.9783 | 0.9963 | 0.9102 | 0.9019 |
| $R^2_{adj}$ | 0.9768 | 0.9960 | 0.9046 | 0.8958 |
| $SD$ (mg g$^{-1}$) | 3.957 | 6.217 | 9.126 | 33.81 |
| General order | | | | |
| $q_n$ (mg g$^{-1}$) | 78.98 | 356.3 | 136.6 | 355.2 |
| $k_n$ (min$^{-1}$ (g mg$^{-1}$)$^{n-1}$) | $1.114 \times 10^{-6}$ | $4.140 \times 10^{-4}$ | $4.964 \times 10^{-7}$ | $2.595 \times 10^{-5}$ |
| n (-) | 22.69 | 2.6270 | 33.08 | 40.08 |
| $R^2$ | 0.9852 | 0.9985 | 0.9629 | 0.9799 |
| $R^2_{adj}$ | 0.9838 | 0.9983 | 0.9580 | 0.9831 |
| $t_{0.5}$ (hour) | 1.46 | 0.24 | 1.57 | 0.43 |
| $T_{0.95}$ (hour) | 6.00 | 2.98 | 6.95 | 3.21 |
| $SD$ (mg g$^{-1}$) | 3.311 | 4.049 | 8.861 | 10.22 |

The general order model had the highest $R^2_{adj}$ and lowest SD values for both dyes on both BBCs (Table 5) and was, therefore, considered as the most suitable model type. The general order kinetic equation gives different values for n (order of adsorption rate) when both dyes—RB-4 and RO-16—concentrations change. Hence, it is hard to make an accurate comparison of the model's kinetic parameters. Therefore, $t_{0.5}$ and $t_{0.95}$ were utilized to compare the RO-16 and RB-4 adsorption kinetics on the ZnCl$_2$-BBC and KOH-BBC carbons. The $t_{0.5}$ and $t_{0.95}$ represent the time (h) when 50% and 95% of saturation ($q_e$) is achieved, respectively [23–25]. For RO-16 on the ZnCl$_2$-BBC and KOH-BBC samples, $t_{0.5}$ was 1.46 and 0.24 h, respectively, while $t_{0.95}$ was 6.00 and 2.98 h. For RB-4 on the ZnCl$_2$-BBC and KOH-BBC samples, $t_{0.5}$ was 1.57 and 0.43 h, respectively, while $t_{0.95}$ was 6.95 and 3.21 h, respectively (Table 5).

Due to the BBCs' textural properties and chemical surface features, the KOH-BBC had faster kinetics compared to ZnCl$_2$-BBC (Table 5), when the values of $t_{0.5}$ and $t_{0.95}$ are considered. KOH-BBC exhibited a much higher $S_{BET}$ and higher amount of micro and mesopores (see Table 2), and this could also be the reason for the better efficiency in the adsorption process. The RB-4 and RO-16 have molecular sizes of 1.59 and 1.68 nm (see Figure 3B), respectively, and are, therefore, readily adsorbed in micro- (<2 nm) and mesopores (2–50 nm). KOH-BBC also has a more hydrophilic surface (Table 2 and Figure 6), which increases the bulk solution's dispersion and the contact between the dyes and available adsorption sites on the KOH-BBC surface.

The adsorption work was further continued by establishing the contact times such as 6.5 and 3.5 h for ZnCl$_2$-BBC and KOH-BBC for RO-16, respectively; and 7.5 and 3.6 h for ZnCl$_2$-BBC and KOH-BBC for RO-16, respectively. The established contact times were slightly higher than the $t_{0.95}$ to ensure that the adsorption process had enough time to reach the equilibrium between the dyes and the BBCs.

*3.3. Equilibrium of Adsorption*

The equilibrium curves and their parameters are shown in Figure 8 and Table 6, respectively.

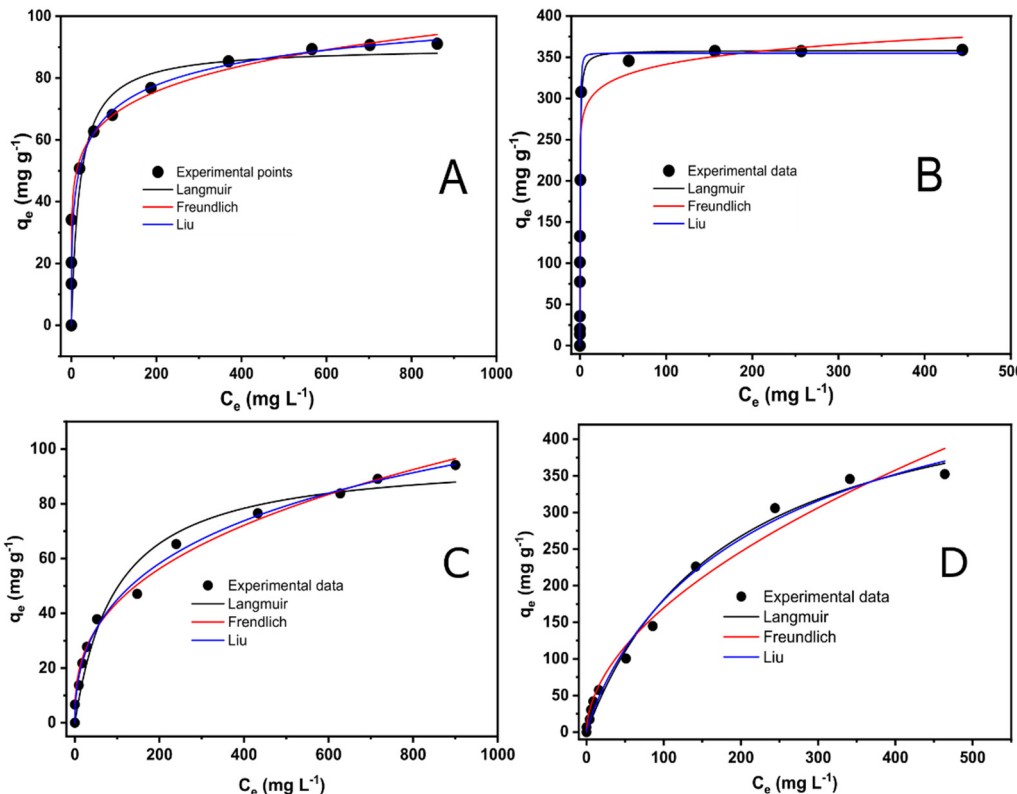

**Figure 8.** Isotherms of adsorption for RO-16 onto ZnCl$_2$-BBC (**A**) and KOH-BBC (**B**) and for RB-4 onto ZnCl$_2$-BBC (**C**) and KOH-BBC (**D**). Contact time 6.5 and 3.5 h for ZnCl$_2$-BBC and KOH-BBC for RO-16, respectively; and 7.5 and 3.6 h for ZnCl$_2$-BBC and KOH-BBC for RO-16, respectively; Initial pH of 5.5 and 4.0 for RO-16 and RB-4, respectively; the adsorbent dosage of 1.5 g L$^{-1}$.

**Table 6.** Equilibrium parameters of RO-16 and RB-4 onto KOH-BBC and ZnCl$_2$-BBC.

| Model | Samples | | | |
|---|---|---|---|---|
| | ZnCl$_2$-BBC | KOH-BBC | ZnCl$_2$-BBC | KOH-BBC |
| Langmuir | **RO-16** | | **RB-4** | |
| Q$_{max}$ (mg g$^{-1}$) | 90.04 | 358.2 | 59.00 | 339.15 |
| k$_L$ (L mg$^{-1}$) | 0.05004 | 2.579 | 0.02698 | 0.005491 |
| $R^2$ | 0.8386 | 0.8614 | 0.9534 | 0.9905 |
| $R^2_{adj}$ | 0.8225 | 0.8488 | 0.9488 | 0.9896 |
| SD (mg g$^{-1}$)$^2$ | 13.77 | 57.72 | 4.794 | 13.93 |
| Freundlich | | | | |
| k$_F$ ((mg g$^{-1}$) (mg L$^{-1}$)$^{-1/nF}$) | 34.37 | 257.1 | 11.10 | 14.25 |
| n$_F$ (dimensionless) | 6.7123 | 16.24 | 3.906 | 1.859 |
| $R^2$ | 0.8480 | 0.8467 | 0.9889 | 0.9818 |
| $R^2_{adj}$ | 0.8328 | 0.8328 | 0.9878 | 0.9799 |
| SD (mg g$^{-1}$)$^2$ | 13.36 | 60.71 | 2.399 | 19.31 |
| Liu | | | | |
| Q$_{max}$ (mg g$^{-1}$) | 123.1 | 354.9 | 332.9 | 582.5 |
| k$_S$ (mg L$^{-1}$) | 0.02017 | 1.943 | 0.007468 | 0.004040 |
| n$_L$ (dimensionless) | 0.3850 | 1.780 | 0.2913 | 0.8848 |
| $R^2$ | 0.8498 | 0.8646 | 0.9891 | 0.9911 |
| $R^2_{adj}$ | 0.8164 | 0.8375 | 0.9899 | 0.9891 |
| SD (mg g$^{-1}$)$^2$ | 13.99 | 59.84 | 2.344 | 14.21 |

For both BBCs and dyes, the Liu isotherm had the best fit. It was, therefore, used to describe the RO-16 and RB-4 removal for both BBCs.

Liu's model assumes that the adsorption has a heterogeneous behavior due to different active sites acting simultaneously and with different free adsorption energies [23,24]. However, a saturation of the adsorbent takes place, attaining the maximum adsorption capacity ($Q_{max}$).

For RO-16 on the $ZnCl_2$-BBC and KOH-BBC samples, $Q_{max}$ was 90.1 and 354.8 mg g$^{-1}$, respectively, while RB-4 was 332.9 and 582.5 mg g$^{-1}$ (Table 6). Thus, the KOH-BBC adsorbed almost three times more RO-16 and 60% more RB-4 than the $ZnCl_2$-BBC. Its higher $S_{BET}$ value and lower hydrophobicity can explain the better performance of KOH-BBC when compared to $ZnCl_2$-BBC, already discussed earlier.

For both BBCs, RB-4 presented higher $Q_{max}$ when compared to RO-16. Both dyes are water-soluble and carry two anionic sulfonic groups in their molecules and remain anionic in aqueous solutions [42]. On the other hand, both BBCs have their surfaces positively charged (see Table 4, pH are 5.1 and 6.0 for $ZnCl_2$-BBC and KOH-BBC samples, respectively). While the adsorption process is happening, the pH of the solution loaded with the BBCs is around 5.8–6.2; this leads to the presence of H$^+$ in the solution, which leads to the protonation of cationic groups such amino groups present on BBCs surfaces [42–44]. This enhances the adsorption of both dyes RO-16 and RB-4 dyes due to electrostatic interactions [42–44].

Additonally, as mentioned in the kinetic discussion, the RB-4′s smaller molecule size may facilitate the diffusion of the RB-4 molecules into the BBC's micro and mesopores.

### 3.4. Mechanism of Adsorption

Taking into account the porosity data such as $S_{BET}$, pore size distribution, HI, the chemical nature of the adsorbents, initial pH solution, kinetics of adsorption, and equilibrium studies result for the RB-4 and RO-16 dyes onto BBCs samples, it is possible to suggest the primary mechanisms of adsorption for both dyes on BBCs (see Figure 9).

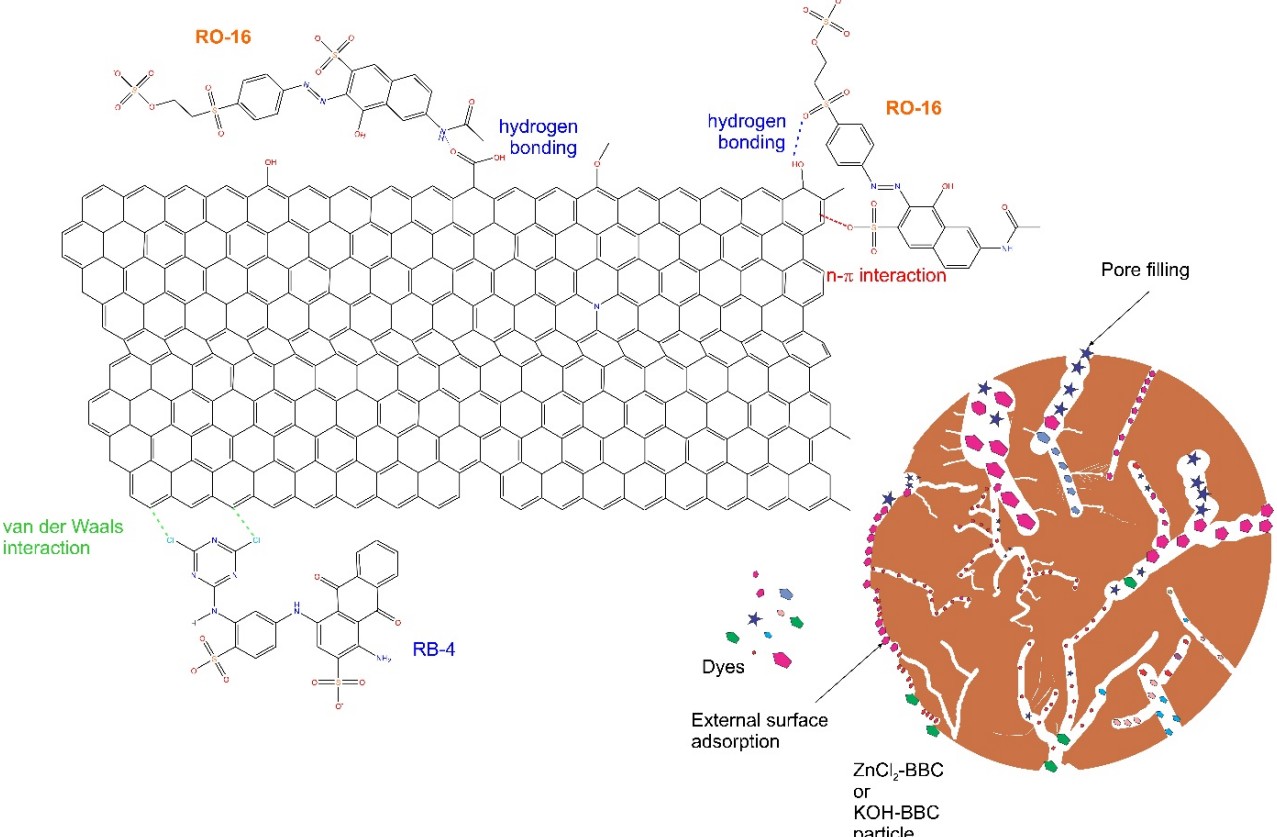

**Figure 9.** Schematic mechanism of adsorption of RO-16 and RB-4 onto BBC structure.

The adsorption process takes place through different physical interactions between BBC surfaces and dyes such as hydrogen bonding, hydrophobic interactions, and $\pi$-$\pi$ and n-$\pi$ interactions of the aromatic ring of the BBCs with the aromatic rings of the dyes [45]. Donor-acceptor interactions (n-$\pi$ interaction) occur among aromatic rings in the BBC structures that act as an electron acceptor (see Figure 9). In addition, the aromatic rings of both RB-4 and RO-16 molecules interact with the C=O, OH, COOH, and phenyl groups of the BBCs that act as adsorption sites (see Figure 9) [45].

Another mechanism that takes place on the RB-4 and RO-16 adsorption process onto BBCs is the pore-filling due to the highly developed porosity and high $S_{BET}$ values. The pore-filling can be the most prominent process that contributes to the high adsorption efficiency for both dyes onto highly porous BBCs (see Figure 9).

*3.5. Adsorbent Performance: Comparison with Literature*

The spruce bark $ZnCl_2$-BBC and KOH-BBC performances were compared with other adsorbents' literature data (Table 7). Assuming that the literature data displays optimized conditions for each BBC, the KOH-BBC is the second most efficient, having the second-highest adsorption capacity ($Q_{max}$) for RO-16 removal and the highest for RB-4.

**Table 7.** Comparison of KOH-BBC and ZnCl2-BBC concerning the reported literature in terms of capacity.

| Adsorbent | pH | Dosage (g $L^{-1}$) | T (°C) | $Q_{max}$ (mg $g^{-1}$) | Ref. |
|---|---|---|---|---|---|
| RO-16 | | | | | |
| BBC-KOH-800 | 5.5 | 1.5 | 22 | 354.8 | This study |
| BBC-ZnCl$_2$-800 | 5.5 | 1.5 | 22 | 90.1 | This study |
| Chitosan/sepiolite composite | 6.5 | 2.0 | 30 | 190.96 | [46] |
| Fish scales Mesoporous BBC | 6.0 | 1.0 | 50 | 114.2 | [47] |
| BBC Brazilian-pine fruit shell | 2.5 | 2.5 | 50 | 314.0 | [48] |
| BBC Brazilian-pine fruit shell | 2.5 | 2.5 | 50 | 470.0 | [48] |
| BBC from rice husk ash | 11 | 2.5 | 30 | 13.32 | [49] |
| Phosphoric BBC from biomass | 6.2 | 0.4 | 30 | 58.54 | [50] |
| Psyllium seed powder biosorbent | 4.0 | 2.0 | 30 | 206.6 | [51] |
| Paper sludge activated carbon | 2.0 | 1.0 | 30 | 178.0 | [52] |
| Ananas Comosus leaves BBC | 2–3 | 1.0 | 30 | 147.05 | [53] |
| Sewage sludge BBC | 2.0 | 10.0 | 25 | 114.7 | [54] |
| Coffee husk-based BBC | 4.0 | 2.0 | 30 | 66.76 | [55] |
| Coffee husk-based BBC | 4.0 | 2.0 | 50 | 76.57 | [55] |
| RB-4 | | | | | |
| BBC-KOH-800 | 4.0 | 1.5 | 22 | 582.5 | This study |
| BBC-ZnCl$_2$-800 | 4.0 | 1.5 | 22 | 332.9 | This study |
| Multi-walled carbon nanotubes | 2.0 | 1.5 | 25 | 502.5 | [42] |
| Single-walled carbon nanotubes | 2.0 | 1.5 | 25 | 567.7 | [42] |
| Chitosan hydrogel beads (CHB) | 4.0 | 1.0 | 30 | 317 | [43] |
| CHB modified with hexadecylamine | 4.0 | 1.0 | 30 | 454 | [43] |
| Enteromorpha prolifera BBC | 6.0 | - | 27 | 131 | [56] |
| Mg–Al layered double hydroxide | 2.0 | 0.75 | 22 | 328 | [57] |
| Cotton grafted with chitosan | 4.0 | 10 | 25 | 180 | [58] |

It is worth highlighting that the spruce bark KOH-BBC's $Q_{max}$ for RB-4 is comparable to that of the single-walled carbon nanotubes studied by Machado et al. [58] (582.5 vs. 567.7 mg $g^{-1}$), but the production cost of carbon nanotubes is substantially higher when compared to KOH-BBC. Additionally, Table 7 shows and compares the spruce bark BBCs with different adsorbents reported in the literature. It is shown that BBC Brazilian-pine fruit shell [47] exhibited the highest $Q_{max}$ for RO-16; however, the adsorption conditions were very different when compared to this work, e.g., the temperature was higher (50 °C vs. 22 °C) as well as the adsorbent dosage (66.6% more adsorbent than was used by this work), which means increasing the costs involved in the adsorption process. This also needs to

be considered when the effectiveness of adsorbent material is evaluated and compared with others.

Thus, it can be concluded that both BBCs (especially KOH-BBC) are suitable adsorbents for the elimination of dyes with competitive and efficient adsorption capacities.

### 3.6. Treatment of Synthetic Dye Effluents

According to the adsorption data (kinetic and equilibrium), both BBCs were very efficient for removing RB-4 and RO-16 from aqueous solutions, indicating that these BBCs could also be employed to treat real effluents. Therefore, both BBCs were tested for the treatment of two synthetic dyeing effluents. The BBCs' removal percentage of dye mixture in the effluents was evaluated from UV–vis spectra of the untreated and treated effluents (see Figure 10).

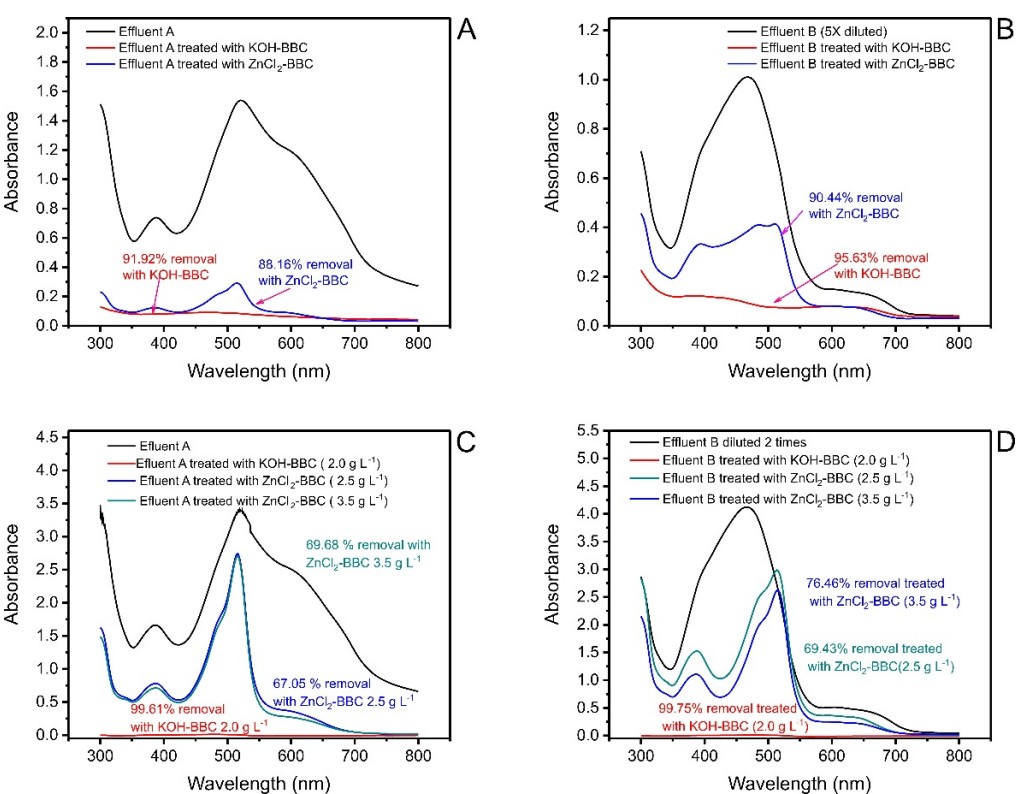

**Figure 10.** Adsorption of synthetic dyes effluent. (**A**) Effluent A; (**B**) Effluent B. (**C**) Effect of BBC mass dosage on effluent A treated and (**D**) effect of BBC mass dosage on effluent B treated.

$ZnCl_2$-BBC removed 88.2% and 90.4% for the effluent A and B, respectively, while KOH-BBC removed 91.9% and 95.6% at an adsorbent dosage of 1.5 g $L^{-1}$ (Figure 10A,B).

With KOH-BBC, only 2.0 g $L^{-1}$ was needed to remove almost 100% of all compounds in both effluents (Figure 10C,D). $ZnCl_2$-BBC removed 69.7% and 76.5% at a dosage of 3.5 g $L^{-1}$. These differences agree with the previously reported adsorption data and discussed in the work where the KOH-BBC had better adsorption properties than $ZnCl_2$-BBC. Still, a good removal percentage was achieved for both BBCs. However, it should point out that the KOH activation could be considered a more interesting method because zinc salts (e.g., $ZnCl_2$) are more expensive and toxic [59] when compared to KOH, which is a corrosive chemical reagent [60]; therefore, it would be preferable to use a cheaper and non-toxic reagent, such as KOH, for BBC preparation.

## 4. Possible Application of Used BBC after Adsorption of Dyes

The re-use or final disposal of the BBC materials loaded with the selected adsorbate is an important question when designing an adsorption system or new adsorbent

materials. BBC can be regenerated and reused many times without losing adsorption performance [7,29]. However, after being fully saturated, its final disposal or other utilization must be considered once they no longer can be regenerated for water treatment application [7]. The main employed methods to manage used BBC are landfill disposal and incineration [23,24]. However, in some cases, used adsorbents are used as soil fertilizer [23,24], depending of the type of the adsorbate loaded on the BBc surface. These methods are influenced by some factors such as, cost of the adsorbent, type and toxicity of the pollutant, costs involved with the methods including the cost of the combustion and incineration plant, and fees for disposal. Although landfills have typically been used for the disposal of sorbents, as well as soil fertilizers, these methods might have subsequent pollution risk when toxic compounds leach from adsorbents into the soil.

## 5. Conclusions

The spruce bark BBCs were produced using $ZnCl_2$ and KOH as the activation agents. The BBC characteristics were strongly dependent on the type of activating agent. KOH-BBC had a higher $S_{BET}$ (1067.2 $m^2$ $g^{-1}$) and a larger pore volume (0.5584 $cm^3$ $g^{-1}$) than $ZnCl_2$-BBC. However, the KOH-BBC had a more developed aromatic structure. KOH treatment generated a BBC with a more well-developed porosity and a higher number of mesopores than $ZnCl_2$-BBC. Additionally, KOH-BBC had a less hydrophobic surface and a higher H and O content than $ZnCl_2$-BBC. However, the carbon yield for KOH-activation was 63% lower than for $ZnCl_2$-activation. For both dyes' adsorption on both BBCs, the general-order model and the Liu model exhibited the best fitness for adsorption kinetics and equilibrium, respectively. The equilibrium $Q_{max}$ at 22 °C was for RO-16 on KOH-BBC and $ZnCl_2$-BBC 354.8 and 90.1 mg $g^{-1}$, respectively, and for RB-4 582.5 and 332.9 mg $g^{-1}$. Based on characterization and experimental data, it was suggested that electrostatic interactions and hydrogen bonds between BBCs and RO-16 and RB-4 dyes played the most important role in the adsorption process. In an analysis of removing two synthetic effluents, both BBCs had good outcomes in the percentage; the BBC made with KOH had much better performances. We have shown that efficient and low-priced BBCs can be produced from Norway spruce bark through simple activation procedures. These results call for further studies on underlying mechanisms and how to optimize the treatment procedures for different applications.

**Author Contributions:** Conceptualization, G.S.d.R.; investigation G.S.d.R.; formal analysis, G.S.d.R. and T.N.P.; data curation, G.S.d.R.; writing—original draft preparation, G.S.d.R.; funding acquisition, S.H.L.; writing—review and editing, S.H.L., M.T., H.P.d.O., E.C.L. and G.L.D. All authors have read and agreed to the published version of the manuscript.

**Funding:** This research was funded by the Treesearch Postdoctoral program, Bio4Energy—a Strategic Research Environment appointed by the Swedish government, and the Swedish University of Agricultural Sciences.

**Institutional Review Board Statement:** Not applicable.

**Informed Consent Statement:** Not applicable.

**Data Availability Statement:** Not applicable.

**Acknowledgments:** Lima thanks CNPq, CAPES, and FAPERGS for supporting his researches. The authors are grateful to ChemAxon for giving us an academic research license for the Marvin Sketch software, Version 21.3.0 (http://www.chemaxon.com), 2021 used for molecule physical-chemical properties.

**Conflicts of Interest:** The authors declare no conflict of interest.

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
