# Peer review of "Preparation and Application of Efficient Biobased Carbon Adsorbents Prepared from Spruce Bark Residues for Efficient Removal of Reactive Dyes and Colors from Synthetic Effluents"

_coatings, doi:10.3390/coatings11070772_

Round 1

Reviewer 1 Report

1. In this work, spruce bark residues were used to produce carbon-based materials (BBCs). The effect of two chemical activators (KOH and ZnCl2) on BBCs was extensively investigated. Interestingly, adsorption experiments were also performed to evaluate the ability of the two BBCs to remove two dyes and treat synthetic effluents. 

We are thankful to the reviewer for all suggestions and recommendations given for helping to improve this manuscript.

2. Although the topic is of great interest, in my opinion, the article would benefit from a revision of the abstract to improve the readability, and both English and syntax should be addressed to improve the manuscript. In general, the abbreviation should be defined in the abstract, and subscripts/superscripts are missing throughout the abstract; please check.

The abstract was checked and improved. See it in the main text.

3. In my opinion, line 20 should be “with ZnCl2 or KOH, and pyrolysis…”.

Dear reviewer, your suggestion was taken into account. See it in the manuscript.

4. In lines 23-24, the authors stated that KOH-BBC has more and larger pores with respect to ZnCl2-BBC. The same concept is repeated in line 26; please check.

The repeated part was removed from line 26.

5. Line 37-38 I would expect this information at the beginning of the abs.

Dear Reviewer, the information was re-written. See it below and in the manuscript.

“The biobased carbon materials showed high efficiency for removing RO-16 and RB-4, comparable to the best examples from the literature. Also, both the KOH- and ZnCl2-BBC showed a high ability to purify two synthetic effluents, but the KOH-BBC was superior.”

6. Line 39 “can be prepared from spruce bark,” same of lines 19-21.

Dear Reviewer, the information was erased. See it in the response above.

7. Line 64-65 biobased carbon materials and active carbon are defined with the same abbreviation; please check.

Dear Reviewer, the abbreviation was updated. (AC)

8. Line 102. Despite the title “Preparation of BBCs and synthetic dyes,” dyes are not mentioned in this paragraph (2.1.1); please modify the title.

It was done.

9. Line 198-200. Why did the authors choose this specific contact time? Have they been optimized? 

Dear Reviewer, the contact time was determined by studying the kinetic experiments. From Table 5, we can see the kinetic parameters and calculated t0.95 (The t0.95 represents the time (h) when 95% of saturation (qe) is achieved). By this value, it is possible to estimate with accuracy the time when the adsorption process is close to reaching the equilibrium process.

Maybe the text was placed in the wrong place, removing it to the Kinetic studies section. (3.2.1. Adsorption Kinetics)

10. Line 200-201. The authors stated that “The extra contact time was to ensure that the contact time at any dye concentrations adsorbate will be enough to reach the equilibrium,” is the extra contact time referred to the ZnCl2-BBC? Why is it necessary only in this case?

Dear reviewer, the extra contact time was imposed for both BBCs. See in the text below and in the manuscript (page 15).

“ The adsorption work was further continued by establishing the contact times such as 6.5 and 3.5 hours for ZnCl2-BBC and KOH-BBC for RO-16, respectively; and 7.5 and 3.6 hours for ZnCl2-BBC and KOH-BBC for RO-16, respectively. The established contact times were slightly higher than the t0.95 to make sure that the adsorption process had enough time to reach the equilibrium between the dyes and the BBCs.”

11. I suggest moving paragraph 2.3.3 before the Dye adsorption analysis.

Dear Reviewer, thanks for your suggestion. However, we understand that it makes more sense to keep it is already given. The synthetic effluent removal tests should be presented and discussed after the dye adsorption.

12. In my opinion, lines 239-250 could be rephrased to avoid repetitions.

Dear Reviewer, thanks for your suggestion. The text was re-written. See below and in the manuscript.

“The SEM images show remarkable differences between both microstructures. The ZnCl2-BBC has a dense structure, with more elongated cavities and holes of different sizes and shapes (2A and 2C) that should have been formed during the leaching step with 6.0 mol L-1 HCl. Also, it is observed that ZnCl2-BBC presents a rough surface.

KOH-BBC (Fig 2B and 2D) present ordered macropore structure and holes with lower diameter in its surface, which should be attributed to the lower concentration of HCl (1.0 mol L-1) used in the leaching step. Both BBC presents irregular particle size and rough surface.”

13. Line 265 Figure 2 should be Figure 3.

It was done.

14. Although I really appreciate the detailed comparison reported in table 3, the preparation conditions should be addressed: verbs are missing, °C, please uniform the sentence.

It was done.

15. Did the authors perform FT-IR analysis also on BBC samples before activation? A comparison with the “control sample” could better highlight the difference between the two treatments.

Dear Reviewer, unfortunately, we do not have the FTIR of the non-activated sample, and right now, we are unable to perform it. However, I do believe that this will not affect the quality of the article even because the FTIR and the other analyses highlighted the differences of both treatments on the BBC structures.

Since the water vapor adsorption data are described in paragraph 3.1.3 (line 362), I suggest reporting here the HI results and its equation. Otherwise, HI abbreviation should be defined in Table 4

16. Line 367 should be Table 4

It was fixed.

17. Line 374-377. Please rephrase.

Dear Reviewer, thanks for your suggestion. The text was re-written. See below and in the manuscript.

“Different on isotherm curves may be because the process is complex and does not depend only on the porosity. The adsorption of water vapor on biomass materials is known to be dependent on surface chemistry. BBC materials have plenty of surface functional groups, which initiate predominant water adsorption through the hydrogen bonding between a water molecule and surface functional groups.”

18. The caption of Figure 7 should be addressed.

In the revised manuscript:

Figure 7. Kinetics of adsorption curves for uptake of RO-16 onto ZnCl2-BBC (A) and uptake of RO-16 onto KOH-BBC (B) uptake of RB-4 onto ZnCl2-BBC (C) uptake of RB-4 onto KOH-BBC (D). initial pH of 5.5 and 4.0 for RO-16 and RB-4, respectively, the adsorbent dosage of 1.5 g L-1. The temperature was 22°C.

19. Line 404 Figure 4 should be Figure 6.

It was fixed.

20. Why do the authors choose the RO-16 and RB-4 dyes?

Dear Reviewer, considering all dyes classes, Reactive Dyes have are highly employed in many textile industries. RO-16 and RB-4 are reactive dyes, and this dye class comprises highly colored organic compounds that have primary application in tinting textiles. Although reactive dyes are primarily employed for dying textiles, 15-50% of them are left hydrolyzed in the water baths during the dying process. Based on that, it is relevant to remove RB-4 and RO-16 dyes from aqueous effluents.

21. Table 6 and 7, “Reactive orange 16” and “Reactive blue 4” should be indicated as RO-16 and RB-4 as in table 5.

It was fixed.

22. Line 426. Qmax abbreviation should be defined here, and it is not necessary to repeat it again in lines 466-467.

It was done.

23. Lines 429-432 the same concept as lines 400-405. Please rephrase.

Dear Reviewer, thanks for your suggestion. The text was re-written. See below and in the manuscript.

Line 400

KOH-BBC exhibited a much higher SBET and higher amount of micro and mesopores (see Table 2), and this could also be the reason for the better efficiency in the adsorption process. The RB-4 and RO-16 have molecular sizes of 1.59 and 1.68 nm (see Fig 3B), respectively, and are, therefore, readily adsorbed in micro- (< 2 nm) and mesopores (2 – 50 nm). KOH-BBC also has a more hydrophilic surface (Table 2 and Figure 6), which increases the bulk solution’s dispersion and the contact between the dyes and available adsorption sites on the KOH-BBC surface.

Line 428

For RO-16 on the ZnCl2-BBC and KOH-BBC samples, Qmax was 90.1 and 354.8 mg g-1, respectively, while it for RB-4 was 332.9 and 582.5 mg g-1 (Table 6). Thus, the KOH-BBC adsorbed almost three times more RO-16 and 60% more RB-4 than the ZnCl2-BBC. Its higher SBET and lower hydrophobicity values can explain the better performance of KOH-BBC when compared to ZnCl2-BBC, already discussed earlier.

24. Line 436. Table 3 should be Table 4.

It was done.

25. The resolution of Figure 9 is poor; please check.

Please see Fig. 9 of the revised manuscript.

26. Line 488. “For the effluent A and B, ZnCl2-BBC removed 88.2% and 90.4%, respectively, while 488 KOH-BBC removed 91.9% and 95.6%”, which concentration of KOH and ZnCl2 was used in these cases?

Dear reviewer, the adsorbent dosage for both BBCs was highlighted. See the text below and in the manuscript.

“For the effluent A and B, ZnCl2-BBC removed 88.2% and 90.4%, respectively, while KOH-BBC removed 91.9% and 95.6% at an adsorbent dosage of 1.5 g L-1 (10A and B).”

27. Since the effluents (A and B) contain different dyes, how do the authors calculate the removal percentage?

Absorbance is an additive property. The UV-Vis spectra before the adsorption correspond to the sum of all the chemicals. After the adsorption, the intensity of the bands decreased. Therefore the overall removal (removal of the total effluent) can be calculated by the area of the spectra after adsorption divided by the area of the spectra prior to treatment. This procedure has been employed with success for the simulated effluents in several papers:

REF 10, 15, 32, 42, 48,

28. References style should be uniformed through all sections.

Minor

Line 1-3 seem to have different font sizes or line spacing.

It was fixed.

Line 137 in should be for.

It was done.

In all equations reported, I suggest the use of x instead of .

Dear reviewer, thanks for your suggestion but we kindly refuse to change it. The use of “.” is also quite common in adsorption equations, and it is mathematically accepted. In addition, x is used for vectors.

Line 226 and 230 superscripts are missing.

It was fixed.

Line 263 of misspelling

It was correct; please see the caption of Figure 2.

Table 7 qmax should be Qmax

It was Done.

Line 484 something is missing (could)

According to the adsorption data (kinetic and equilibrium), both BBCs were very efficient for removing RB-4 and RO-16 from aqueous solutions, indicating that these BBCs could also be employed to treat real effluents.

Figure 10 treatment should be treated.

Figure 10. Adsorption of synthetic dyes effluent. A) Effluent A; B) Effluent B. C) effect of BBC mass dosage on effluent A treated and D) ) effect of BBC mass dosage on effluent B treated.

Reviewer 2 Report

The manuscript “Preparation and application of efficient biobased carbon adsorbents prepared from spruce bark residues for efficient removal 3 of reactive dyes and colors from synthetic effluents” report a detailed and interesting investigation on the possibility of producing biobased carbon materials for adsorption properties, starting from an abundant yet widely unused waste such as spruce bark. The Authors precisely detail the production procedure and well characterize the obtained BBC. Overall this is a nice work and is worthy of publication in Coatings. However, the authors need to provide just minor revisions to the manuscript before publication to address a few typos and add some information that might benefit the reader. The specific comments are provided below.

1. Page 2 line 57/58: “and pulp industries 57 drive its utilization are driven.” Please re-check the sentence for redundancy

Dear reviewer, your suggestion was taken into account. See it below and in the manuscript.

“The Swedish annual forest harvest amounts to approximately 90 Mm3 standing volume [5], and they are economically very important for sawmill and paper and pulp industries.”

2. Page 2 line 59: change “the only stem wood is used£ into “only the stem wood is used”

Dear reviewer, your suggestion was taken into account. See it below and in the manuscript.

“However, in the production of sawn timber, pulp, and paper, only the stem wood is used - the remaining components can be considered industrial by-products.”

3. Page 2, line 88: change “consommes” into “consumes.”

It was done.

4. In the Materials And Methods section, since an actual Materials paragraph is missing, the Authors should add some more information (some of which is cited later on in the paper) about RB-4 and RO—16 the first time they cite them. Moreover, since they comment a lot later on about the interaction of such molecules with the different substrates, the molecule drawing would very much help the reader.

Dear reviewer, thanks for your suggestion but we believe that the dye molecules are already shown in the Figure 9. Besides the very useful information are already written in the main text. Besides, this paper has already 10 figures and we believe that this is enough and all important information is already given. We also believe that the readers won’t have any problem to understand the paper’s concept and outcomes.

5. In Eq 1, is what is Ce: this entity is not defined

It was fixed. Cf instead Ce.

6. Page 5, lines 208/209: Authors refer to Supplementary Information, but the reviewer could not retrieve them: please amend.

Dear reviewer, there is no supplementary material. The sentence was re-written.

“Detailed information about all these equations can be found in the literature [8-10, 23, 24].”

7. Figure 3A: please modify X-axis to start from 0.

It was done.

8. Page 9, line 274 remove the dash after micropore

It was done.

9. Page 9 line285: Authors refer to a peak in Fig. 3B; please re-check

Dear reviewer, the phrase was re-written. See it below and in the manuscript.

“BBC-KOH showed a much higher distribution of the pores in the range of large micropores or small mesopores.”

10. Page 10 line 295: Please check the consistency of this sentence: “ …washing, where after the carbonized BBC was mixed with KOH”

Dear reviewer, the phrase was re-written. See it below and in the manuscript.

“First, pre-carbonization at 500°C for one hour followed by NaOH washing; Secondly, BBC was mixed with KOH at a ratio of 1:4 (biomass: KOH) and pyrolyzed at 800°C for 30 min and then, followed by HCl washing to remove the inorganic compounds.”

11. Page 11 In Table 4, the Authors report the BBC yields: do they refer to the starting bark feed or the “bark+activator” mixture. Please define, maybe even in the experimental section.

The yield (%) was calculated from the dry matter quota of the biomass precursor after and before activation.

12. Page 14, line 372: remove the full stop after out

Dear reviewer, see it below and in the manuscript.

“The N2 and H2O isotherms differ both in type and shape. Although there is a non-existing correlation between these two techniques, it is worthwhile to point out that N2 adsorption generated type I isotherms, while H2O adsorption yields isotherms of type V.”

13. Page 15 Table 5: adjust T95 to t0.95

It was done.

14. Page 17, line 449: what do the Authors mean with “The adsorption process is potentialized through” please clarify the sentence

Dear reviewer, see it below and in the manuscript.

“The adsorption process takes place through different physical interactions between BBC surfaces and dyes such as hydrogen bonding, hydrophobic interactions, and π-π and n-π interactions of the aromatic ring of the BBCs with the aromatic rings of the dyes.”

Reviewer 3 Report

1. Lines 44-48.  Since this article is devoted to the adsorption of dyes, such areas of use of biomass as the production of adsorbents for the adsorption of carbon dioxide and as energy source are irrelevant; therefore these areas should be excluded or replaced by the application areas related to the article topic.

Dear reviewer, nothing was said about the adsorption of carbon dioxide. Lines 44-48 is an introductory sentence highlighting that the use of biomass as the carbon source to produce bio-based carbon materials is a sustainable application that helps to reduce CO2 emissions. Nothing was said about the application of BBC in CO2 adsorption.

2. The bark is a well-known raw material for charcoals, and therefore the introduction must be supplemented with literary sources on the production of activated charcoals from the bark of spruce and other trees, the properties of these charcoals.

Dear reviewer, according to the best of our knowledge, few works used Norway spruce bark as the main precursor for the preparation of BBC. Both are mentioned below and in the main text. However, the examples are exhibited in Table 3, “Comparison of KOH-BBC and ZnCl2-BBC preparation methods and SBET for a variety of biomass precursors.” Also, a text was added in page 9, see it below.

“Literature data reveals significant variance in SBET values depending on the type of biomass and preparation conditions (Table 3). For instance, Sipola et al. [8], prepared activated carbon from scots pine (Pinus sylvestrus) and spruce (Picea spp.) barks for wastewater purification and found specific surface areas ranging from 200 to 600 m2 g-1. In another work [9], the spruce bark porous materials were produced and employed in methylene blue dye adsorption. The materials presented SBET ranging from 351 to 1275 m2 g-1 and were successfully employed in the dye removal from aqueous solutions.”

3. The results discuss various characteristics of BBC, such as external surface area, percentages of mesopore and mesopore areas, t-plot micropore area and volume, volume of mesopores, total pore volume and average pore size.  Therefore, in 2.2, the methods of determinations of these characteristics should be described

The specific surface area was calculated in the relative pressure interval of 0.05–0.3 using the Brunauer– Emmett–Teller (BET) method [28]. Mesopore size and distribution were calculated by the Barrett–Joyner–Halenda (BJH) method from desorption curves while the micropore area values were calculated by the t-plot method [28]. The percentage of the mesopore and micropore areas were calculated based on the SBET values [22].

5. Figure 2. Since it is difficult to find the difference between images A and B at low magnifications, it is better to remove these images, and use only the images C and D taken at higher magnifications.

Dear Reviewer, the differences are highlighted in the text. Besides, keeping the images does not compromise the quality of the paper and I believe the images help the readers to better visualize and understand the BBC morphologies.  

6. Table 4. To prove the chemical purity of the adsorbents, the ash content also must be indicated in this table.

Dear reviewer, your suggestion was taken into consideration. See Table 4.

7. Line 324: “…in the BBCs' graphene layers”. Remark: Since graphene layer is monomolecular layer of carbon, which is absent in bulk carbon materials, the word “graphene” should be replaced with word “graphite”.

Done.

8. The 3.1.3 section, Figure 6.  The sorption-desorption hysteresis indicates capillary condensation of water vapor in the mesopores. However, there are no specifications of this process. Therefore, the text of  3.1.3 section should be supplemented by calculations of mesopores average size and total mesopore volume.

Dear reviewer, the accurate calculations regarding the mesopores' average size and total mesopore volume are described in 3.1.1. Textural properties and porosity and Table 2. The results were based on N2 adsorption-desorption. The idea to show Fig. 6 (water vapor adsorption) was to investigate and try to correlate the BBC surface and hydrophobicity properties. It makes no sense to calculate the mesoporosity based on these curves. Therefore, your suggestion was not taken into account.

Additional General Remarks

(1). It should be noted in the discussion that zinc salts are expensive and toxic; therefore, it is preferable to use a cheaper and non-toxic reagent, KOH, for activation of carbon.

Dear reviewer, your suggestion was taken into consideration. A sentence highlighting the higher toxicity of ZnCl2 was added in the main text (lines 487 – 495). See the text below too.

“With KOH-BBC, only 2.0 g L-1 was needed to remove almost 100% of all compounds in both effluents (Figure C and D). ZnCl2-BBC removed 69.7% and 76.5% at a dosage of 3.5 g L-1. These differences agree with the previously reported adsorption data and discussed in the work where the KOH-BBC had better adsorption properties than ZnCl2-BBC. Still, a good removal percentage was achieved for both BBCs. However, it should point out that the KOH activation could be considered a more interesting method because zinc salts (e.g., ZnCl2) are more expensive and toxic [59] when compared to KOH, which is a corrosive chemical reagent [60]; therefore, it would be preferable to use a cheaper and non-toxic reagent, such as KOH, for BBC preparation. “

Added reference.

59. Plum, L.M.; Rink, L.; Haase, H. The Essential Toxin: Impact of Zinc on Human Health. Int. J. Environ. Res. Public Health 2010, 7, 1342-1365.

(2). It is necessary to discuss how to dispose and recycle the used BBC adsorbents after adsorption of dyes: incinerate them? or use the other methods?

Possible application of used BBC after adsorption of dyes

The re-use or final disposal of the BBC materials loaded with the selected adsorbate is an important question when designing an adsorption system or new adsorbent materials. BBC can be regenerated and reused many times without losing adsorption performance [7,29]. However, after being fully saturated its final disposal or other utilization must be considered once no longer they cannot be regenerated for water treatment application [7]. The main employed methods to manage used BBC are landfill disposal and incineration [23,24]. However, in some cases used adsorbents are used as soil fertilizer [23,24], depending of the type of the adsorbate loaded on the BBc surface.  These methods are influenced by some factors such as, cost of the adsorbent, type and toxicity of the pollutant, costs involved with the methods including the cost of the combustion and incineration plant, and fees for disposal. Although landfills have typically been used for the disposal of sorbents, as well as soil fertilizers, these methods might have subsequent pollution risk when toxic compounds leach from adsorbents into the soil.

Round 2

Reviewer 3 Report

Since the authors revised the manuscript taking into account the comments of the reviewer, I can now recommend the revised paper for publication.